# UBQLN Family Members Regulate MYC in Lung Adenocarcinoma Cells

**DOI:** 10.3390/cancers15133389

**Published:** 2023-06-28

**Authors:** Parag P. Shah, Levi J. Beverly

**Affiliations:** 1James Graham Brown Cancer Center, University of Louisville, Louisville, KY 40202, USA; ppshah04@louisville.edu; 2Department of Medicine, Division of Hematology and Oncology, University of Louisville School of Medicine, Louisville, KY 40202, USA

**Keywords:** UBQLN1, UBQLN2, EMT, c-MYC, metastasis

## Abstract

**Simple Summary:**

The ubiquilin family (UBQLN) of proteins consists of five closely related members (UBQLN1, UBQLN2, UBQLN3, UBQLN4, and UBQLNL). The role of UBQLN1 and UBQLN2 in regulating processes involved in cancer progression and tumorigenesis is still not completely understood. MYC is an oncogene and is well known to play important roles in cancer progression and metastasis. Herein, we show that loss of UBQLN1/UBQLN2 causes increased cell viability, cell proliferation, cell migration, clonogenic potential, and cell cycle progression, which is associated with increased MYC expression and overexpression of UBQLN1 reverse the increased expression of MYC following the loss of UBQLN2. Finally, we show that loss of UBQLN1 drives tumorigenesis and lung metastasis in mice which are associated with an increase in the expression of MYC and proteins involved in cell cycle progression. Taken together, our results suggest for the first time a novel role of UBQLN1 and UBQLN2 in regulating MYC in lung adenocarcinoma cells.

**Abstract:**

The ubiquilin family (UBQLN) of proteins consists of five closely related members (UBQLN1, UBQLN2, UBQLN3, UBQLN4, and UBQLNL) that have a high degree of similarity at the level of both amino acid and domain structure. The role of UBQLN1 and UBQLN2 in regulating processes involved in cancer progression and tumorigenesis is still not completely understood. MYC is an oncogene and is well known to play important roles in cancer progression and metastasis. Herein, we show that the loss of UBQLN1 and UBQLN2 causes increased cell viability, cell proliferation, cell migration, clonogenic potential, and cell cycle progression, which is associated with increased MYC expression. UBQLN1 and UBQLN2 interact with phosphorylated MYC and facilitate its degradation. The overexpression of UBQLN1 reverses the increased expression of MYC following the loss of UBQLN2. Further, we present evidence that decreasing MYC levels back to baseline can reverse phenotypes driven by the loss of UBQLN1 or UBQLN2. Finally, we show that loss of UBQLN1 drives tumorigenesis and lung metastasis in mice which are associated with an increase in the expression of MYC, proteins involved in cell cycle progression, and EMT. Taken together, our results suggest for the first time a novel role of UBQLN1 and UBQLN2 in regulating MYC in lung adenocarcinoma cells.

## 1. Introduction

Ubiquilins (UBQLN) belong to a family of UBL/UBA proteins that have been shown to be associated with the ubiquitin–proteasome system [1,2]. The UBQLN family of proteins comprises five protein family members UBQLN1-4 and UBQLNL. It has been reported that the deregulation of UBQLN1 and/or UBQLN2 is associated with various neurological disorders [3,4,5]; however, their cellular and biochemical role in various cell types including lung cancer remain largely unexplored. Previous work from our lab reported that, apart from their role in the endoplasmic reticulum (ER)-associated degradation (ERAD), they play an important role in epithelial-mesenchymal transition (EMT) in lung adenocarcinoma cells. Moreover, we reported that UBQLN1 is frequently lost and underexpressed in lung cancer cell lines as well as human lung adenocarcinomas [6].

Lung cancer is the second most diagnosed cancer in both men and women in the United States and is the leading cause of cancer death. Lung cancer is a multistep process that involves genetic alterations and activation of growth-promoting pathways. A detailed understanding of multiple signaling pathways involved in the pathogenesis of lung cancer is crucial to the development of better treatment strategies. Most commonly, MYC proto-oncogenes are overexpressed in cancer patients and have been intensively investigated as potential targets of cancer treatment [7,8]. Furthermore, MYC is involved in cell growth, differentiation, cell survival, and death [9,10,11,12].

In the present study, we provided evidence that UBQLN1 and UBQLN2 regulate MYC stability in lung adenocarcinoma cells. We show that the loss of UBQLN1 and/or UBQLN2 increases cell viability, cell proliferation, clonogenic potential, and cell migration, which is associated with an increase in MYC expression. We also observed the activation of MYC transcriptional programs, along with several proteins involved in cell cycle progression including cyclin B1, E1, CDK2, and CDK6 following the loss of UBQLN1 and/or UBQLN2. By performing immunoprecipitation experiments, we observed UBQLN1 interaction with the phosphorylated form of MYC (pS62). Interestingly, we showed that restoring MYC back to baseline levels can block increased cell viability, cell migration, and clonogenic potential induced by the loss of UBQLN1 and UBQLN2. Finally, we showed that the loss of UBQLN1 drives tumorigenesis and lung metastasis in mice, which was associated with an increase in MYC expression and proteins involved in cell cycle progression. These data demonstrate that UBQLN1 and UBQLN2 regulate MYC in lung adenocarcinoma cells.

## 2. Materials and Methods

### 2.1. Antibodies Used for the Study

UBQLN1 #14526, UBQLN2 #85509, Cyclin D1 #2978, Cyclin D3 #2936, Cyclin A2 #4656, Cyclin E1 #4129, Cyclin E2 #4132, Cyclin H #2927, Cyclin B1 #4138, CDK2 #2546, CDK4 #2906, CDK6 #3136, GAPDH #5174, c-MYC #9402, p-MYC #13748, MYCBP #517020, VDAC # 4866, CREB #9197, Cdc42 #2462, E-cadherin #3195, N-cadherin #13116, Snail #3879, Slug #9585, Zeb1 #3396, β-catenin #8480, Claudin1 #4933, GFP #2956, (Cell Signaling Technologies Inc. Danvers, MA, USA); Tubulin #B512, FLAG M2 conjugated agarose beads, FLAG poly-clonal #F7425, FLAG Peptide #F-3290 (Sigma-Aldrich, Inc. St. Louis, MO, USA); UBQLN3#376548, UBQLN4 #136145, β-actin #517582 (Santa Cruz Biotechnology, Inc. Dallas, TX, USA); Alexa Fluor 488 goat anti-rabbit IgG #A11034 (Molecular Probes, Invitrogen detection technologies, Eugene, OR, USA); Alexa Fluor 568 Phalloidin #A12380 (Life technologies, Eugene, OR, USA). Ubqln polyclonal was made by inoculating rabbits with a peptide specific to Ubqln1 (Yenzym Antibodies LLC, Brisbane, CA, USA).

### 2.2. Cell Culture and siRNA Transfection and Protein Analysis

Human lung adenocarcinoma cell lines A549, HOP62, H23, and H2009 were purchased from American Type Culture Collection (ATCC, Rockville, MD, USA) and cultured in RPMI medium supplemented with 10% fetal bovine serum (Invitrogen, Carlsbad, CA, USA) and 1% antibiotic/antimycotic (Sigma, St Louis, MO, USA). Human Embryonic Kidney cells (293T; ATCC^®^ CRL3216TM) were cultured in DMEM medium supplemented with 10% fetal bovine serum (Invitrogen, Carlsbad, CA, USA) and 1% antibiotic/antimycotic (Sigma, St Louis, MO, USA). The cell lines were routinely subcultured every 3–5 days. All siRNA transfections were performed using Dharmafect1 # T-2001-03 (Thermo Fisher Scientific Inc, Pittsburgh, PA, USA) as per the manufacturer’s protocol. After a total of 72 h of transfection cells were harvested in CEB lysis buffer # FNN0011 (Invitrogen, Life Technologies, Grand Island, NY, USA, 14072). Protein was quantitated by using Pierce’s BCA Protein Assay Reagent Kit (#23227) from Pierce Biotechnology, Rockford, IL, USA as per the manufacturer’s protocol.

### 2.3. siRNA Sequences Used for the Study

All siRNAs used for the study were ordered from Thermo Fisher Scientific Biosciences Inc., Lafayette, CO, USA.
   Non targeting siRNA   (siNT): UAAGGCUAUGAAGAGAUACAA   UBQLN1 siRNA   (siU1): GAAGAAAUCUCUAAACGUUUUUU
   (siU1-2): GUACUACUGCGCCAAAUUU   UBQLN2 siRNA   (siU2-5): CCUGGUAUCUCUAAGUAUAUU
   (siU2-6): GUAGAAUCUGAGUGUAAUAUU   UBQLN3 siRNA   (siU3): Cat. No. L-013398-00   UBQLN4 siRNA   (siU4): Cat. No. L-021178-01   Kif11 siRNA   (siKif11): Cat. No. L-003317-00   MYC siRNA   (siMYC): GGACUAUCCUGCUGCCAAGUU

### 2.4. Cell Viability/Cell Proliferation Assay

Cell viability/cell proliferation assay was performed by using alamarBlue™ (#R7017)) from Sigma-Aldrich, Inc., St. Louis, MO, USA as per the manufacturer’s protocol. Briefly, cells were transfected either with non-targeting siRNA (siNT) or with two different siRNAs targeting UBQLN1 (siU1 and siU1-2) and two different siRNAs targeting UBQLN2 (siU2-5 and siU2-6). After 24 h of transfection, cells were trypsinized and 1000 cells were reseeded in 96-well plates; cell viability was assessed for 5 consecutive days using alamarBlue. Similarly, cell viability was performed following either loss of MYC or UBQLN1 or combined loss of both MYC and UBQLN1 with indicated siRNA concentration.

### 2.5. Flow Cytometry Analysis

Flow cytometry analysis was performed by using FACScan Flow Cytometer from Cytek Development, Fremont, CA. USA. For the BrdU cell proliferation assay, A549 cells were transfected either with non-targeting siRNA (siNT) or with two different siRNAs targeting UBQLN1 (siU1 and siU1-2) and two different siRNAs targeting UBQLN2 (siU2-5 and siU2-6). After 48 h of transfection, cells were incubated with the anti-BrdU antibody for one hour. After subsequent washing, cells were incubated with 7AAD and analyzed by flow cytometry analysis. For apoptosis assay, A549 cells were prepared as described earlier followed by incubation with AnnexinV antibody for one hour. After subsequent washing, cells were incubated with 7AAD and analyzed for apoptosis by flow cytometry analysis. Data were analyzed with FlowJo software version 10.

### 2.6. Cell Migration Assay or Scratch Assay or Wound Healing Assay

Cell migration assay or Scratch assay was performed as described previously [6]. Briefly, lung adenocarcinoma cells A549 and HOP62 were plated on 6-well plates in triplicates. Cells were transfected either with non-targeting siRNA (si_NT) or with siRNA targeting UBQLN1 (si_U1), siRNA targeting UBQLN2 (si_U2-5), siRNA targeting MYC (si_MYC) and combination of siRNA targeting UBQLN1 and UBQLN2 with MYC (si_U1+ si_MYC and si_U2-5+ si_MYC). After 24 h of siRNA transfections, the wound was made using the pipette tip following replacement with fresh media. Cells were examined successively after 24 h and 48 h of wound formation and photographed. Quantification of relative percentage wound closure was performed by using ImageJ software version 1.53t.

### 2.7. Immunoprecipitation

DNA transfections in human embryonic 293T cells were performed using PEI (poly-ethylamine) (#764965) from Sigma-Aldrich, Inc., (St. Louis, MO, USA) as per the manufacturer’s protocol. Cell extracts were prepared following scrape harvesting of 293T cells using CHAPS lysis buffer (1% CHAPS detergent, 150 mM NaCl, 50 mM Tris pH 7, 5 mM EDTA), as per the manufacturer’s protocol. Immunoprecipitations were performed with anti-FLAG conjugated agarose beads. Briefly, 400 μg of protein was incubated in 400 uL of total CHAPS buffer and incubated with anti-FLAG conjugated agarose beads for 2 h at 4 °C. Following incubation and successive washing with CHAPS buffer, SDS loading buffer was added directly to wash matrix, boiled, and loaded directly into the wells of a PAGE gel. Similarly, for detecting endogenous interaction of UBQLN1 with p-MYC following siRNA transfections, total 1 mg of protein lysate was used to pull down UBQLN1 with UBQLN1 antibody. Lysates were further incubated with protein A beads for 2 h at 4 °C. Following incubation and successive washing with CHAPS buffer, SDS loading buffer was added directly to wash matrix, boiled, and loaded directly into the wells of a PAGE gel.

### 2.8. Western Blot Analysis

Cells were lysed in Cell Extraction buffer (CEB #FNN0011) from Invitrogen Corporation, Camarillo, CA. USA, supplemented with Complete Mini Protease Inhibitor and phosphatase inhibitor tablets (Roche Molecular Biochemical, Indianapolis, IN, USA). Protein quantification was performed by using PierceTM BCA protein assay (#23228, #1859078) from Fisher Scientific, Rockford, IL, USA, according to the manufacturer’s instruction. An equal amount of protein (40 µg) from each sample was added to the SDS loading buffer, boiled, and resolved on a 4–12% SDS–PAGE gel and transferred onto a membrane. Blots were probed with different antibodies and immunoreactive target proteins were visualized using the SuperSignal West Femto Maximum Sensitivity Substrate (#PI34095) from Fisher Scientific (Pittsburgh, PA, USA), according to the manufacturer’s instruction. The membrane was stripped by using a western blot stripping reagent (BioRad, Hercules, CA, USA) and reprobed with GAPDH or tubulin or β-actin antibody to normalize the variation in the loading of samples.

## 3. Results

### 3.1. Loss of UBQLN1 or UBQLN2 Increases Cell Proliferation, Cell Cycle Progression, and Clonogenic Potential in Lung Adenocarcinoma Cells

In the present study, we explored the effect of the knockdown of UBQLN1 and/or UBQLN2 on cell proliferation and clonogenic potential in lung adenocarcinoma cell lines. Interestingly, we observed an increase in relative cell numbers following a loss of either UBQLN1 or UBQLN2 (Figure 1A). To determine whether increased cell numbers were due to either decreased apoptosis or increased cell cycle progression, we performed annexin/7AAD staining and BrdU labeling. Following siRNA-mediated knockdown and staining cells with annexin/7AAD, we performed flow cytometry analysis. Interestingly, we did not observe any significant change in apoptosis following the loss of either UBQLN1 or UBQLN2 (Figure 1(Bi,Bii)). As we observed a drastic increase in relative cell numbers (Figure 1A) following the siRNA-mediated knockdown of UBQLN1 or UBQLN2, we next performed a BrdU cell proliferation assay. Measuring the cell cycle by BrdU revealed a significant increase in the S-phase following the loss of UBQLN1 or UBQLN2 (Figure 1(Ci,Cii)). To further confirm our BrdU results, we performed a western blot analysis of proteins involved in cell cycle progression following the loss of UBQLN1 or UBQLN2. Interestingly, using lung adenocarcinoma cell lines including A549, HOP62, and H23, we observed an increased expression of cdk2, 4, and 6 following the loss of either UBQLN1 or UBQLN2 (Figure 1D). We also observed a drastic increase in the expression of cyclin D1 and D3 involved in cell cycle progression following the loss of UBQLN1 and UBQLN2 (Figure 1D). Next, we were interested to see the effect of UBQLN1 and UBQLN2 knockdown on clonogenic potential and cell proliferation in lung adenocarcinoma cells. We use two different siRNA targeting UBQLN1 (si_U1 and si_U1-2) and UBQLN2 (si_U2-5 and si_U2-6). We observed a significant increase in clonogenic potential following the loss of either UBQLN1 or UBQLN2 (Figure 1(Ei,Eii)).

Loss of UBQLN1 or UBQLN2 increases the clonogenic potential and activates proteins involved in cell cycle progression in HPL1D cells.

Earlier we observed an increase in cell viability and cell proliferation following the loss of either UBQLN1 or UBQLN2 in lung adenocarcinoma cells (Figure 1A,(Ci,Cii)). This increase in cell viability and proliferation was associated with an increase in clonogenic potential in lung adenocarcinoma cells including A549, H2030, and H2009 (Figure 1(Ei,Eii)). We also observed an increase in the expression of proteins involved in cell cycle progression including cdks and cyclins (Figure 1D). Next, we were interested to see whether the phenotype we observed following the knockdown of either UBQLN1 or UBQLN2 is cancer specific. We extended our findings and performed similar experiments in transformed human peripheral lung epithelial cell line HPL1D. Interestingly, we observed similarities in results with lung adenocarcinoma cell lines. We observed an increase in clonogenic potential in HPL1D cells following either loss of UBQLN1 or UBQLN2 (Appendix A). Furthermore, consistent with lung adenocarcinoma cell lines, we also observed an increase in the expression of proteins involved in cell cycle progression including CyclinD3 and CyclinB1 following the loss of UBQLN1 (Appendix A). We also observed an increase in the expression of MYC-binding protein mycBP following the loss of UBQLN1. These findings demonstrate that the phenotype we observed following the loss of UBQLN is not cancer-specific (Appendix A).

The loss of UBQLN1 or UBQLN2 increases cell viability and clonogenic potential in lung adenocarcinoma cells H2030, H2009, and A549 cells.

Previously, we observed that the loss of UBQLN1 or UBQLN2 increases cell viability and clonogenic potential in H2030, H2009, and A549 cells (Figure 1A,(Ei,Eii)). We further confirmed the knockdown of UBQLN1 and UBQLN2 by performing western blot analysis (Appendix A). Before staining the colonies at the end of the 10th day, we assess the cell viability by performing a cell viability assay using alamarBlue. We observed a significant increase in cell viability following the loss of UBQLN1 and UBQLN2 (Appendix A).

### 3.2. Loss of UBQLN1 or UBQLN2 Activates an MYC Transcriptional Program through Increased MYC Nuclear Localization and Expression

Previously, we used microarray analysis to identify signaling pathways altered following the loss of UBQLN1 or UBQLN2. To determine if the loss of UBQLN1 or UBQLN2 could alter MYC protein levels, we performed western blot analysis following the loss of UBQLN1, UBQLN2, UBQLN3, or UBQLN4 in A549 and HOP62 cell lines. Interestingly, we observed an increase in the expression of MYC following the loss of UBQLN1 and UBQLN2 but not the loss of UBQLN3 or UBQLN4 (Figure 2A). Next, we performed immunofluorescence staining of MYC in lung adenocarcinoma cell lines following a loss of UBQLN1 or UBQLN2. Immunofluorescence staining of MYC revealed a drastic increase in nuclear expression of MYC following the loss of UBQLN1 and/or UBQLN2 in lung adenocarcinoma cells (Figure 2B,C).

### 3.3. Loss of UBQLN1 and/or UBQLN2 Stabilizes MYC Protein

The MYC transcription factor is involved in the regulation of cell proliferation, apoptosis, and differentiation [12,13]. Aberrant increased MYC expression has been reported in many human cancers including colon, breast, and lung cancer [14] even though it has an extremely short half-life (∼30 min) [15]. Furthermore, it has been shown that rapid turnover of MYC is required for normal growth control. We observed an increase in the expression of MYC following the loss of UBQLN1 or UBQLN2 in lung adenocarcinoma cell lines including A549, HOP62, H23, and H2009 (Figure 3A). Next, we were interested to explore MYC stability following the loss of UBQLN1 and/or UBQLN2. We performed the siRNA-mediated knockdown of UBQLN1 in A549 cells followed by treatment with cycloheximide for different time intervals (30 min to 120 min). We observed that UBQLN1 loss in A549 stabilizes MYC protein levels (Figure 3B). Next, we were interested to see the effect of the treatment of cycloheximide on the stability of MYC following the loss of both UBQLN1 and UBQLN2. We performed the siRNA-mediated knockdown of UBQLN1 or UBQLN2 or a combination of both UBQLN1 and UBQLN2 in A549 and HOP62 cells followed by cycloheximide treatment. Interestingly, we observed similar results following the loss of both UBQLN1 and UBQLN2 (Figure 3C). We further confirmed our findings of stabilization of MYC following the loss of UBQLN1 in additional lung adenocarcinoma cell lines PC9 and H358 (Figure 3D).

### 3.4. UBQLN1 Interacts with MYC Phosphorylated on S62

The ubiquitin–proteasome pathway is a highly specific system for the selective degradation of short-lived proteins such as MYC [16]. We observed an increase in the expression of MYC following the loss of UBQLN1 in the lung adenocarcinoma cell line. Next, we were interested to see whether UBQLN1 directly interacts with the phosphorylated form of MYC. We transfected 293T cells with either empty vector or with FLAG-tagged UBQLN1-wt followed by immunoprecipitation with anti-FLAG conjugated agarose beads. Interestingly, we observed that UBQLN1 interacts with the phosphorylated form of MYC (on serine 62) (Figure 4B). The MYC expression is highly controlled at several levels, including transcription, translation, and post-translation (protein stability) [17]. Sears et al. showed that phosphorylation of c-Myc at conserved residues serine 62 (S62) and threonine 58 (T58) can regulate c-Myc protein stability in response to mitogen signaling [18]. Popov and coworkers showed that ubiquitylation regulates MYC and ubiquitylation of the amino terminus of MYC by SCF (beta-TrCP) antagonizes SCF(Fbw7)-mediated turnover [19]. MYC pathogenicity generally results from aberrant levels of expression, resulting from increased transcription, chromosomal amplification, or rearrangement [18]. MYC protein levels present in the cells are regulated by the proteasomal pathway. Since we saw changes in MYC stability and the phosphorylation of MYC is known to alter protein degradation, we were interested to see whether, by inhibiting proteasome, we could alter the stability of MYC. We transfected 293T cells with either empty vector or with UBQLN1-wt and treated cells with either vehicle alone or with MG132 for indicated time points before immunoprecipitation with anti-FLAG conjugated agarose beads. We observed that MG132 treatment does not alter the extent of interaction between UBQLN1 and the phosphorylated form of MYC (Figure 4B). As we observed how UBQLN1 interacts with the phosphorylated form of MYC, we were interested to see which domain of UBQLN1 is responsible for its interaction with p-MYC. Using our engineered UBQLN1 constructs (Figure 4A), we performed an immunoprecipitation experiment in 293T cells. Interestingly, we observed that the UBL domain of UBQLN1 is responsible for its interaction with p-MYC (Figure 4C). Next, we were interested to see in which sub-cellular compartment UBQLN1 and p-MYC occur. We performed subcellular fractionation following transfection of either empty vector (EV) or UBQLN1 in 293T cells. Western blot analysis revealed that UBQLN1 co-fractionates with p-MYC in the nucleus, albeit at low levels (Figure 4D). We observed an increase in the expression of MYC following the siRNA-mediated loss of UBQLN1 or UBQLN2. Further, we were interested to see whether an increased expression of UBQLN1 can reverse this increase in the expression of MYC. Interestingly, we observed that an increased expression of MYC following the loss of UBQLN2 was completely abolished by overexpressing UBQLN1 in A549 cells (Figure 4E). Further confirming these findings, we observed an increased interaction of p-MYC with UBQLN1 following the overexpression of UBQLN1 in A549 cells (Figure 4F). Finally, we confirmed our findings of the interaction of p-MYC with UBQLN1 by performing immunoprecipitation using endogenously expressed UBQLN1 and P-MYC in A549 cells. Western blot analysis revealed the endogenous interaction of UBQLN1 with p-MYC using A549 cells (Figure 4G).

### 3.5. Partial Loss of MYC Reverse Increased in Cell Viability and Clonogenic Potential Induced by Loss of UBQLN1

We observed an increase in cell viability and clonogenic potential following the loss of either UBQLN1 or UBQLN2 in lung adenocarcinoma cells, which we hypothesized was due, at least in part, to increased MYC expression. To explore this possibility, we performed siRNA-mediated knockdown of MYC. As expected, the complete loss of MYC almost completely inhibited the growth of cancer cells. However, interestingly, MYC knockdown to basal levels reversed the increase in cell viability following the loss of UBQLN1 (Figure 5A). Furthermore, the partial loss of MYC, which brings MYC levels back to baseline, could reverse the increase in clonogenic potential following the loss of UBQLN1 (Figure 5B,C). We further confirmed the knockdown of UBQLN1 and MYC by performing a western blot analysis (Figure 5D). These results demonstrate a critical balance between UBQLN1 and MYC driving cell viability and clonogenic potential in lung adenocarcinoma cells.

### 3.6. Partial Loss of MYC Reverse Increased Cell Migration Induced by Loss of UBQLN1 and/or UBQLN2

The mesenchymal phenotype is characterized by an increase in migratory and invasive properties [19,20,21,22]. In our study, involving lung adenocarcinoma cell lines, including A549 and HOP62, we observed that the loss of UBQLN1 and UBQLN2 increases cell migration. We hypothesize that an increase in cell migration is associated with an increase in the expression of MYC. We were interested to see whether the partial loss of MYC could reverse cell migration following the loss of UBQLN1 and UBQLN2. We performed a cell migration assay following a combined siRNA-mediated knockdown of either UBQLN1 or UBQLN2 with MYC in lung adenocarcinoma cell lines A549 and HOP62. Interestingly, we observed partial loss of MYC-reversed cell migration following the loss of UBQLN1 or UBQLN2 alone (Figure 6(Ai,Aii),(Bi,Bii)). These results further confirm the critical balance between UBQLN1 or UBQLN2 and MYC driving cell migration in lung adenocarcinoma cells. We further confirmed our finding that the loss of UBQLN1 drives cell migration in lung adenocarcinoma cells due to cell mobility and not due to cell proliferation by performing a single-cell mobility assay using a Keyence live cell imager. Following siRNA transfections either with non-targeting siRNA or with two different siRNA targeting UBQLN1 (siU1 and siU1-2), cell mobility was analyzed for 24 h. We observed a significant increase in cell mobility following the loss of UBQLN1 when compared to cells transfected with non-targeting siRNA (Figure 6(Ci)). Interestingly, when we analyzed cells for differences in proliferation, we did not see any significant difference following the loss of UBQLN1 (Figure 6(Cii)).

### 3.7. Loss of UBQLN1 Induces Tumorigenesis and Lung Metastasis in Mice by Increasing Expression of MYC and Cell Cycle Progression

Using lung adenocarcinoma cell lines, we observed an increase in cell viability, clonogenic potential, and cell migration following the loss of either UBQLN1 or UBQLN2, which was associated with an increase in MYC expression. Next, we were interested to see whether our in vitro findings can be recapitulated in vivo. To explore this possibility, using lentiviral infections, we generated A549 cells stably expressing either UBQLN1 shRNA or control shRNA. Knockdown of UBQLN1 in generated A549 cell line was confirmed by performing a western blot analysis (Figure 7A). Following subcutaneous inoculations of generated cell lines, we observed an increase in tumor volume (Figure 7B,C) and lung metastasis in mice injected with A549 cells expressing UBQLN1 shRNA compared to mice injected with A549 cells expressing control shRNA (Figure 7D). Western blot analysis further confirmed our in vitro findings of an increase in tumorigenesis and lung metastasis in mice injected with A549 cells expressing UBQLN1 shRNA is associated with an increase in MYC expression and proteins involved in cell cycle progression (Figure 7E). We further investigated expression levels of key proteins including E-cadherin, Claudin1, N-cadherin, Slug, Zeb1, and β-catenin, which are known for their important role in the regulation of EMT. We observed a decrease in the expression of epithelial markers including E-cadherin, Claudin-1, and an increase in the expression of mesenchymal markers including N-cadherin, Snail, Slug, Zeb1, and β-catenin following the loss of UBQLN1. These in vivo findings further confirmed that the loss of UBQLN1 drives EMT in lung adenocarcinoma (Figure 7E).

## 4. Discussion

Ubiquilins (Ubqlns) belong to a family of UBL/UBA proteins that are associated with the ubiquitin–proteasome system [2]. Ubiquilin family proteins consist of five protein members (UBQLN1-4, UBQLN-L) which are all closely related. Previous studies from different groups have reported that UBQLN1 and UBQLN2 have been associated with various neurological disorders (Zeng [3,5,23]); however, their cellular and biochemical roles in various cell types including lung cancer remain unexplored. Previous work from our lab reported that, apart from their role in ERAD, they play an important role in the induction of epithelial-mesenchymal transition (EMT) in lung adenocarcinoma cells. Moreover, we reported that UBQLN1 is frequently lost and underexpressed in lung cancer cell lines as well as human lung adenocarcinomas [6].

In the present study, we observed an increase in cell viability following the loss of UBQLN1 and UBQLN2 (Figure 1A) in lung adenocarcinoma cell lines, which was associated with a decrease in early apoptosis as revealed by annexin/7AAD staining (Figure 1B). This increase in cell viability was further confirmed by performing a BrdU cell proliferation assay. Measuring cell cycle phases by BrdU revealed an increase in S-phase following the loss of UBQLN1 or UBQLN2 (Figure 1C). The progression through the cell cycle phases (G1, S, G2, and M) is under the control of a family of serine/threonine protein kinases including the catalytic subunit, the cyclin-dependent protein kinase (Cdk), and a regulatory subunit, the cyclin. Although cdks are inactive, the binding of a cyclin activates them, allowing them to modify target proteins [24]. As MYC expression is tightly correlated with cell proliferation and cell cycle progression and we observed an increase in S-phase following the loss of UBQLN1 or UBQLN2, we were interested to see the effects of UBQLN1 and UBQLN2 knockdown on the expression of cdks and cyclins. Interestingly, we observed the increased expression of cdk 2, 4, and 6 and cyclin D1, D3, and E1 following the loss of UBQLN1 and UBQLN2 in lung adenocarcinoma cells (Figure 1D). Consistent with these findings, we also observed an increased expression of cyclin D3 and B1 in normal peripheral airway HPL1D cells following the loss of UBQLN1 (Appendix A). Since we observed an increase in cell proliferation following the loss of UBQLN1 and UBQLN2, we performed a clonogenic assay in lung adenocarcinoma cell lines following the loss of UQLN1 and UBQLN2. Interestingly, we observed that the loss of UBQLN1 and UBQLN2 increases the clonogenic potential in H2030, H2009, and A549 cells (Figure 1(Ei,Eii)). Furthermore, we also observed an increase in clonogenic potential following the loss of UBQLN1 and UBQLN2 in HPL1D cells, further confirming that the phenotype that we observed following the loss of UBQLN is not as cancer-specific (Appendix A).

We performed microarray analysis to explore which genes were significantly altered following the loss of UBQLN1. Interestingly, we observed that MYC was significantly upregulated following the loss of UBQLN1 in lung adenocarcinoma cell lines. MYC is dysregulated and overexpressed in most human cancers [25]. Several signaling pathways and key regulatory proteins including MYC are associated with cancer progression and metastasis [12,26]. MYC initiates tumorigenesis, which results from tumor-intrinsic mechanisms that regulate different cellular processes, including cell proliferation and apoptosis. Next, we were interested to see whether MYC expression is altered following the loss of other UBQLN family members, including UBQLN3 and UBQLN4. We performed western blot analysis following the loss of UBQLN1, UBQLN2, UBQLN3, or UBQLN4. Interestingly, we observed an increase in the expression of oncogene MYC following the loss of UBQLN1 or UBQLN2, but not with UBQLN3 or UBQLN4 (Figure 2A) in lung adenocarcinoma cells. We further confirmed our findings of an increase in the expression of MYC following the loss of UBQLN1 or UBQLN2 by performing immunofluorescence staining. Immunofluorescence staining of MYC revealed a drastic increase in nuclear expression following the loss of UBQLN1 or UBQLN2, or a combined loss of both UBQLN1 and UBQLN2 in lung adenocarcinoma cells (Figure 2B,C).

MYC is a highly unstable protein and is usually degraded in approximately 30 min in cells [27]. Furthermore, it has been shown that the deregulated reduction in MYC protein degradation results in the accumulation of MYC in many cancers, which may be associated with uncontrolled cell proliferation [28]. Neidler and coworkers in their study showed MYC accelerates KRasG12D-driven lung adenocarcinoma development [29]. In our study using lung adenocarcinoma cell lines, we observed an increase in the expression of MYC following the loss of either UBQLN1 and/or UBQLN2. Next, to determine whether protein synthesis is responsible for increased MYC levels following the loss of UBQLN1 or UBQLN2, we treated lung adenocarcinoma cell lines with protein synthesis inhibitor cycloheximide. As expected, we observed that cells depleted with UBQLN1 or UBQLN2 stabilize MYC compared with the cells transfected with non-targeting controls (Figure 3B–D).

To explore possible mechanisms involved in MYC activation and associated signaling responsible for driving processes, including cell proliferation and clonogenic potential following the loss of UBQLN1 and UBQLN2, we performed immunoprecipitation experiments. Interestingly, we observed that UBQLN1 interacts with MYC phosphorylated on serine 62 (Figure 4B). MYC protein levels present in the cells are regulated by the proteasomal pathway. We were interested to see whether, by inhibiting proteasome, we could alter the activation of MYC. Interestingly, we observed that MG132 treatment does not alter the extent of interaction between UBQLN1 and the activated form of MYC (Figure 4B). Previous findings from our lab reported that UBQLN1 interacts with transmembrane proteins including BCLb [30,31], ESYT2 [31], IGF1R [32], and INSR, and IGF2R interacts with UBQLN1 through its STI-1 and STI-2 domains and are stabilized as a result of this interaction. Next, we were interested to see which domain of UBQLN1 is responsible for the interaction with the phosphorylated form of MYC. Using our engineered constructs (Figure 4A), we performed immunoprecipitation experiments and observed that the UBL domain of UBQLN1 is responsible for driving interaction with the phosphorylated form of MYC (Figure 4C). Furthermore, by performing subcellular fractionation, we showed that UBQLN1 interacts with MYC in nuclear fraction and the overexpression of UBQLN1 completely abolishes the expression of MYC in A549 cells (Figure 4D,E). We showed that the overexpression of UBQLN1 enhances the interaction of UBQLN1 with p-MYC in A549 cells (Figure 4F). We further confirmed the interaction of p-MYC with UBQLN1 by performing immunoprecipitation using endogenously expressed UBQLN1 and P-MYC in A549 cells (Figure 4G). These findings suggest that there exists an interplay between UBQLN proteins and MYC driving cellular processes involved in lung cancer progression.

During EMT, cells lose their epithelial properties and primary site and start translocating to the secondary site where they colonize and ultimately metastasized [33,34]. In the present study, we hypothesized that an increase in the expression of MYC following the loss of UBQLN1 and UBQLN2 is associated with an increase in cell proliferation, cell viability, clonogenic potential, and cell migration. To answer this, we performed a rescue experiment. Interestingly, with partial loss of MYC, we were able to reverse the phenotype that we observed in lung adenocarcinoma cells following the loss of UBQLN1, including cell viability (Figure 5A), clonogenic potential (Figure 5B,C), and cell migration (Figure 6(Ai,Aii),(Bi,Bii)) associated with EMT. Consistent with our findings, recently Sabit and coworkers showed that the knockdown of MYC controls the proliferation of oral squamous cell carcinoma cells in vitro by regulating key apoptotic marker genes [35]. These results clearly demonstrate a critical balance between UBQLN1 and MYC driving cellular processes including cell viability, clonogenic potential, and cell migration in lung adenocarcinoma cells. It has been shown that MYC directly regulates genes involved in cell migration, invasion [36], cell cycle progression [37], and tumor microenvironment leading to metastasis [38]. In the present study, we showed that the loss of UBQLN1 drives tumorigenesis and lung cancer metastasis, which was associated with an increase in MYC expression, cell cycle progression, and EMT (Figure 7B–E).

## 5. Conclusions

In conclusion, the present study provided evidence that UBQLN family members, especially UBQLN1 and UBQLN2, regulate MYC in lung adenocarcinoma cells. We demonstrate that there is a critical balance between UBQLN1 and MYC, driving cellular processes including cell viability, cell proliferation, cell migration, and clonogenic potential in lung adenocarcinoma cells (Appendix A). The findings from the present study suggest that MYC can be a potential target in lung cancer with underexpressed UBQLNs. Future work will explore the exact molecular mechanisms by which UBQLN proteins suppress biological processes including proliferation, migration, and cancer progression through the regulation of MYC.

## Figures and Tables

**Figure 1 cancers-15-03389-f001:**
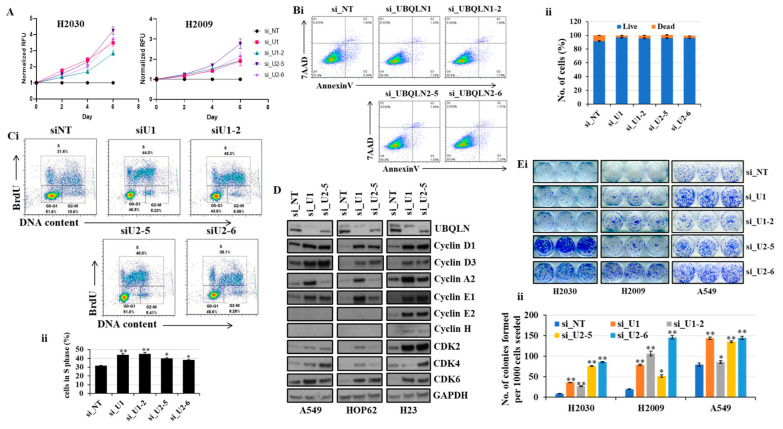
Loss of UBQLN1 and UBQLN2 increases cell viability, cell proliferation, cell cycle progression, and clonogenic potential in lung adenocarcinoma cells. (**A**) Cell viability assay in H2030 and H2009 cells. Cells were transfected as indicated. After 24 h of transfection, cells were trypsinized and 1000 cells were reseeded in 96-well plates and cell viability was assessed for 5 consecutive days. (**Bi**,**ii**) AnnexinV and 7AAD staining in A549 cells. Cells were prepared as described in methodology followed by incubation with AnnexinV antibody for one hour. After subsequent washing, cells were incubated with 7AAD and analyzed for apoptosis by flow cytometry analysis. (**Ci**,**ii**) Cell proliferation assay in A549 cells. Cells were prepared as described in methodology followed by incubation with anti-BrdU antibody for one hour. After subsequent washing, cells were incubated with 7AAD and analyzed by flow cytometry analysis. (**D**) Western blot analysis of cell cycle progression markers in A549, HOP62, and H23 cells. (**Ei**,**ii**) Clonogenic assay in H2030, H2009, and A549 cells (* *p* < 0.05, ** *p* < 0.01). The original western blots is Appendix A.

**Figure 2 cancers-15-03389-f002:**
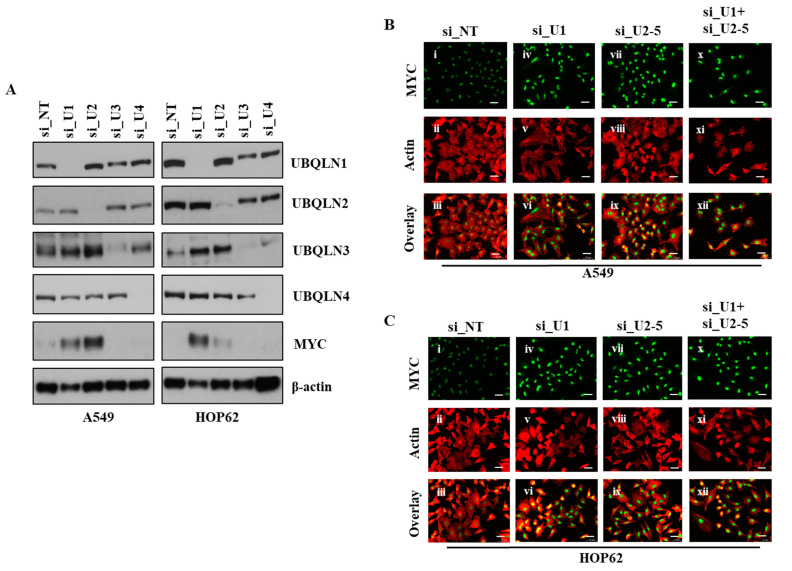
Loss of UBQLN1 and UBQLN2 increases MYC activity and expression of MYC. (**A**) Western blot analysis of UBQLNs and MYC in A549 and HOP62 cells. (**B**,**C**) Fluorescence staining for MYC in A549 and HOP62 cells. After 24 h of transfection either with non-targeting siRNA (si_NT) or with siRNAs targeting UBQLN1 (si_U1), siRNA targeting UBQLN2 (si_U2-5) or a combination of siRNA targeting UBQLN1 and UBQLN2 (si_U1+si_U2-5), cells were trypsinized and plated on chamber slides and stained for MYC. i, iv, vii and x: MYC was detected using Alexa Fluor 488 goat anti-rabbit IgG (green). ii, v, viii and xi: F-actin was stained with Alexa Fluor 568 Phalloidin (red), scale bar: 50 µm. iii, vi, ix and xii: overlay of respective MYC and F-actin staining with DAPI counterstain. The original western blots is Appendix A.

**Figure 3 cancers-15-03389-f003:**
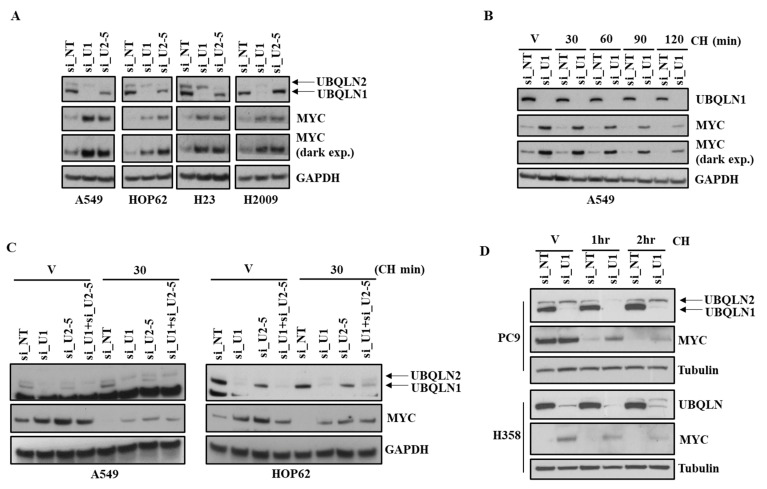
Loss of UBQLN1 and/or UBQLN2 stabilizes MYC. (**A**) Western blot analysis of MYC in A549, HOP62, H23, and H2009 cells. (**B**) Western blot analysis of MYC in A549 cells. Cells were prepared as described in methodology, followed by treatment with either vehicle alone or with cycloheximide for indicated time points. After 72 h of transfection, cells were harvested and analyzed for MYC expression. (**C**) Western blot analysis of MYC in A549 and HOP62 cells. Cells were prepared as described in methodology, except cells were also transfected with combination of siRNA targeting UBQLN1 along with siRNA targeting UBQLN2 (si_U1+si_U2-5). (**D**) Western blot analysis of MYC in PC9 and H358 cells. The original western blots is Appendix A.

**Figure 4 cancers-15-03389-f004:**
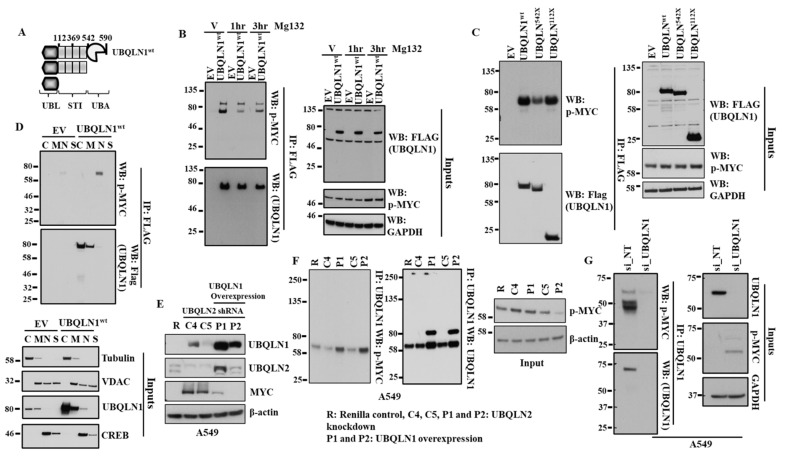
UBQLN1 interacts with p-MYC and treatment with proteasome inhibitor MG132 does not alter p-MYC expression. (**A**) Schematic representation of the engineered UBQLN1 constructs. UBQLN1wt-unaltered UBQLN1 cDNA, UBQLN1542X–UBQLN1 lacking the UBA domain, UBQLN1112X–UBQLN1 encoding only the UBL domain. (**B**) Western blot analysis of UBQLN1 interaction with p-MYC. (**C**) Western blot analysis of UBQLN1 mutant interaction with p-MYC using engineered UBQLN1 mutant that lacks specific domains. (**D**) Western blot analysis of subcellular fractionation to determine location of UBQLN1: p-MYC interaction. Cellular fractions cytosol (C), membrane (M), nuclear (N), and structural/cytoskeletal fraction (S). (**E**) Representative western blot analysis A549 cells confirming overexpression of UBQLN1 blocks increase in expression of MYC following loss of UBQLN2. A549 cells stably expressing either control shRNA (R), UBQLN2 shRNA (C4 and C5), or cells expressing UBQLN2 shRNA and overexpressing UBQLN1 (P1 and P2) were generated and used for analysis of protein expression. (**F**) Representative western blot analysis confirming overexpression of UBQLN1 increases its interaction with p-MYC. (**G**) Representative western blot analysis confirming endogenous interaction of UBQLN1 with p-MYC using A549 cells. The original western blots is Appendix A.

**Figure 5 cancers-15-03389-f005:**
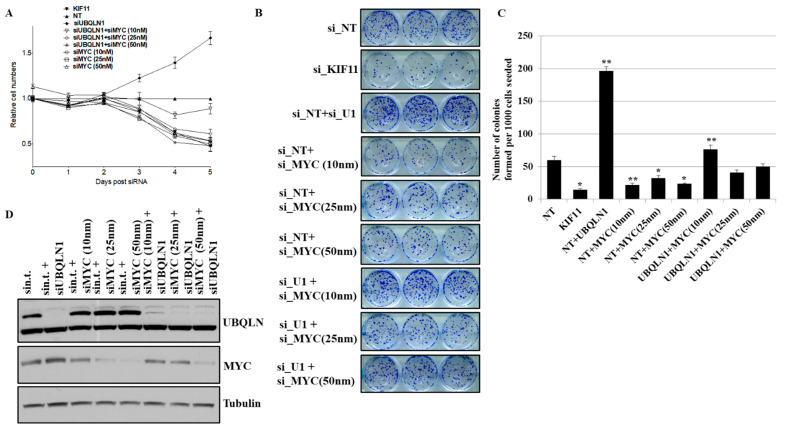
Partial loss of MYC reverse increased in cell viability and clonogenic potential following loss of UBQLN1. (**A**) Cell viability assay in A549 cells. (**B**) Clonogenic potential in A549 cells. Relative number of colonies formed per 1000 cells seeded were counted. (**C**) Quantification of relative number of colonies formed per 1000 number of cells seeded in B (* *p* < 0.05, ** *p* < 0.01). (**D**) Western blot analysis of MYC confirming knockdown in A549 cells. The original western blots is Appendix A.

**Figure 6 cancers-15-03389-f006:**
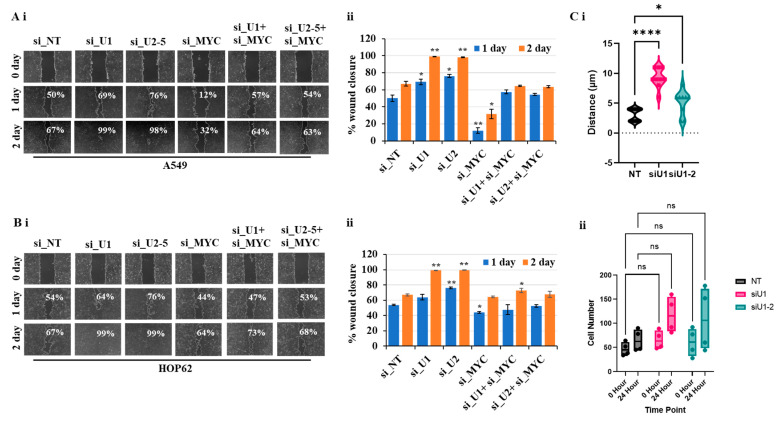
Partial loss of MYC reverses increased cell migration following loss of UBQLN1 and UBQLN2. (**Ai**,**Bi**). Scratch assay in A549 and HOP62 cells. (**Aii**,**Bii**) Bar diagram on the right indicates relative number of migrated cells. (* *p* < 0.05, ** *p* < 0.01, **** *p* < 0.0001, ns: non-significant). (**Ci,ii**). Single-cell mobility and proliferation assay in A549 cells.

**Figure 7 cancers-15-03389-f007:**
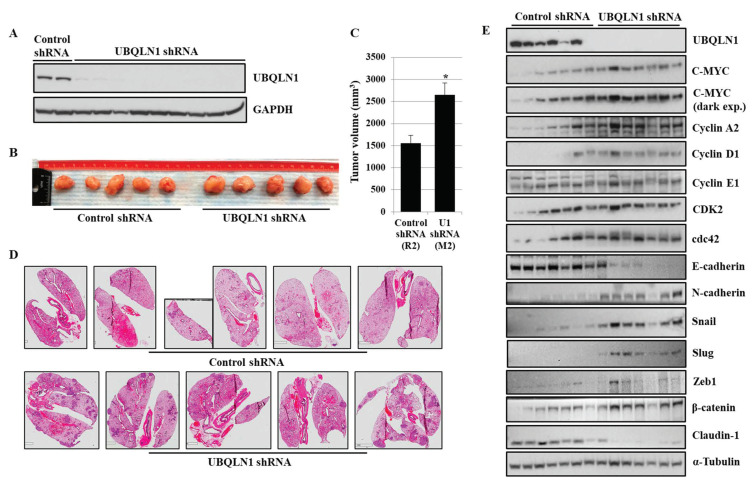
Loss of UBQLN1 induces tumorigenesis and lung metastasis in mice by increasing expression of MYC and cell cycle progression. (**A**) Western blot analysis of UBQLN1 confirming shRNA mediated UBQLN1 knockdown in A549 cells. (**B**) Representative tumors developed by being subcutaneously injected with A549 cells expressing either control shRNA or cells expressing UBQLN1 shRNA. (**C**) Bar diagram comparing tumor volume of generated tumors in mice (* *p* < 0.05). (**D**) H&E staining of paraffin embedded lung sections from mice subcutaneously injected with A549 cells expressing either control shRNA or cells expressing UBQLN1 shRNA. (**E**) Representative western blot analysis of UBQLN1, C-MYC, cell cycle progression markers, and EMT markers on subcutaneous tumors. The original western blots is Appendix A.

## Data Availability

The data presented in this study are available in this article (and Appendix A).

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
