# Peer review of "UBQLN Family Members Regulate MYC in Lung Adenocarcinoma Cells"

_cancers, 2023, doi:10.3390/cancers15133389_

Round 1

Reviewer 1 Report

The manuscript by Shah & Beverly identifies an apparently novel for for UBQLN1 and 2 in restricting proliferation of lung cancer cells via regulation of MYC protein stability.  The manuscript is well written (with some minor alterations needed for clarity) and nicely presented, however some of the interpretations are unsupported by the data provided, use of statistics to support their conclusions is inadequate and the manuscript is poorly referenced, relying predominantly on citations of many rather old reviews.

Specific points that should be addressed:

1) Throughout the manuscript please provide statistical analyses of data reproducibility - this pertains to FACS, AlalmarBlue and Western blotting experiments shown.

2) Some of the figures inserted as landscape rather than portrait have been truncated (Fig S2, Fig 5), obscuring data.

3) Panel B in Figure 3 should be withdrawn and it fails to show reduced MYC in the control setting in response to Cyclohexamide treatment of any of the cell lines shown - this is in contrast with panels C & D where the expected reduction of MYC is indeed observed in the control lanes.

4) The overexpression of proteins in 293T cells in a poor way to demonstrate protein protein interactions and can produce artifacts arising from aggregation of unfolded overexpressed proteins.  Can they demonstrate interaction of endogenously expressed MYC and UBQLN or perhaps use a proximity ligation assay to more convincingly demonstrate the interaction.

5) The cell migration data could simply arise from the effects on cell proliferation and the conclusions drawn from the data shown are unsound.  Use of quantitative time-lapse video microscopy is needed to properly address this aspect of the work.

6) Relevant literature on MYC should be cited:  the work of Popov and Eilers for regulation of MYC by Ubiquitylation, Rosalie Sears work on MYC phosphorylation, Kruspig and Murphy for the metastatic potential of MYC in lung cancer

1) The authors repeatedly refer to an increase in "viability" throughout the manuscript. Given that their analysis in Section 3 Line 177 detected no impact of UBQLN depletion on apoptosis, use of this term is incorrect and should be replaced with increased proliferation and/or clonogenicity as appropriate.

2) Use of the term "rescue" is confusing and should be replaced with "reversed".

Author Response

We are thankful to reviewers for putting their time and effort towards enhancement of quality of manuscript. We really appreciate for the positive feedback regarding our manuscript and the work we carried out within the manuscript. According to reviewer’s suggestions, we have made numerous changes to the text within the manuscript and added additional experimental data further clarifying reviewer’s comments.

Please find below a comprehensive point-by-point clarification of all reviewer comments.

Reviewer 1

Comments and Suggestions for Authors

The manuscript by Shah & Beverly identifies an apparently novel for UBQLN1 and 2 in restricting proliferation of lung cancer cells via regulation of MYC protein stability.  The manuscript is well written (with some minor alterations needed for clarity) and nicely presented, however some of the interpretations are unsupported by the data provided, use of statistics to support their conclusions is inadequate and the manuscript is poorly referenced, relying predominantly on citations of many rather old reviews.

Specific points that should be addressed:

  • Throughout the manuscript, please provide statistical analyses of data reproducibility – this pertains to FACS, AlamarBlue and Western blotting experiments shown.

According to reviewer’s suggestion, statistical analysis of data pertaining to FACS analysis, Alamar Blue assay and clonogenic potential has been incorporated in manuscript and suitable text has been incorporated in text in more detail. Each experiment was repeated at least three times and representative western images has been included in the manuscript. The original uncropped images for each and every western blot analysis has been provided as a as a supplemental information.

2) Some of the figures inserted as landscape rather than portrait have been truncated (Fig S2, Fig 5), obscuring data.

According to reviewer’s suggestion, figures have now inserted in proper format in the manuscript.

3) Panel B in Figure 3 should be withdrawn, and it fails to show reduced MYC in the control setting in response to Cycloheximide treatment of any of the cell lines shown – this is in contrast with panels C & D where the expected reduction of MYC is indeed observed in the control lanes.

We completely agree with reviewer’s suggestion regarding Panel B in Figure 3. Necessary changes have been now incorporated in Figure 3. 

4) The overexpression of proteins in 293T cells in a poor way to demonstrate protein-protein interactions and can produce artifacts arising from aggregation of unfolded overexpressed proteins.  Can they demonstrate interaction of endogenously expressed MYC and UBQLN or perhaps use a proximity ligation assay to more convincingly demonstrate the interaction.

We completely agree with reviewer’s concern regarding MYC and UBQLN interaction using 293T cells. Overexpression proteins in 293T cells is a poor way to demonstrate protein-protein interactions and can produce artifacts arising from aggregation of unfolded overexpressed proteins. According to reviewer’s suggestion, we performed pull down assay with endogenously expressed MYC and UBQLN1 using A549 cells. Interestingly, we observed interaction of p-MYC and UBQLN1 in A549 cells. More convincingly, we did not see any kind of interaction in the A549 cells depleted with UBQLN1 (Figure 4G).

5) The cell migration data could simply arise from the effects on cell proliferation and the conclusions drawn from the data shown are unsound.  Use of quantitative time-lapse video microscopy is needed to properly address this aspect of the work.

We completely agree with reviewer’s concern regarding cell migration assay. The cell migration data could simply arise from the effects on cell proliferation and definitive conclusions cannot be provided. According to reviewer’s suggestion, we performed single cell time lapse mobility assay for 24 hrs using Keyence live cell imager. Interestingly, we observed significant increase in cell mobility following loss of UBQLN1 compared to cells transfected with non-targeting control siRNA. We further noticed that there was not significant increase in cell proliferation. This data clearly demonstrates that increase in cell migration in case lung adenocarcinoma cells following loss of UBQLN1 was due to increase in cell mobility and not due to increase in cell proliferation.  

6) Relevant literature on MYC should be cited:  the work of Popov and Eilers for regulation of MYC by Ubiquitylation, Rosalie Sears work on MYC phosphorylation, Kruspig and Murphy for the metastatic potential of MYC in lung cancer

According to reviewer’s suggestion, relevant literature on MYC has been cited and necessary changes has been incorporated in manuscript. 

 Comments on the Quality of English Language

  • The authors repeatedly refer to an increase in “viability” throughout the manuscript. Given that their analysis in Section 3 Line 177 detected no impact of UBQLN depletion on apoptosis, use of this term is incorrect and should be replaced with increased proliferation and/or clonogenicity as appropriate.

We apologize for using word viability. According to reviewer’s suggestion appropriate changes has been now incorporated in the manuscript.

2) Use of the term “rescue” is confusing and should be replaced with “reversed”.

According to reviewer’s suggestion the term “rescue” has been now replaced with “reversed” and necessary changes has been now incorporated in the manuscript.

Reviewer 2 Report

In this manuscript, the authors investigated the effect of UBQLN on tumor growth and metastasis, and found that UBQLN regulates MYC levels in lung adenocarcinoma cells. It is an interesting topic to study, but the manuscript is poorly organized and riddled with mistakes, and requires substantial proof-reading and reformatting. Some figures are cut off. 

1. Please provide methods for clonogenic assays, immunofluorescence microscopy, xenograft study in mice, and statistical analysis.

2. Figure 1A: Why did the NT control cells barely grow during the 5-day experiment period? Also, it would be great if the curves/datapoints can be made more distinguishable between conditions (i.e. different colors). Right now it is hard to tell them apart.

3. Figure 1C: Are the changes in cell cycle statistically significant? Maybe a histogram would be a better way to show the difference.

4. Figure 1D: Please specify which band is UBQLN1 or UBQLN2.

5. Figure 1E: Please provide statistics for the differences in colony number. Also, does the difference in clonogenic potential between the two different siRNAs targeting the same gene correlate with the knockdown efficiency?

6. Figure S1A and S1B: siU2-6 has similar knockdown efficiency as siU2-5. Why did cells treated with siU2-6 form much fewer colonies than ones treated with siU2-5?

7. Figure S1C: It would be helpful to show the levels of these proteins in si_U2 cells as well.

8. Figure S2: The figure is cut off. Also, in S2Ai, there are 6 lanes on the western blot, but only 5 lanes are labeled.

9. Line 248 and 252: The text referred to the wrong figures.

10. Line 257: Figure S2Aiii does not exist.

11. Figure 2A: Please provide statistics and enrichment score.

12. Figure 3: There seemed to be some delay in MYC degradation in Figure 3A, but it is hard to tell the kinetics in the remaining panels. Is there just more MYC to begin with? It would be important to calculate/measuring the actual half-life of MYC upon UBQLN knockdown.

13. Figure 4C: Does UBA also promote the interaction with MYC since the protein lacking UBA pulled down less MYC? Also, it would be helpful to do the same experiment with a mutant protein lacking UBL to show whether UBL is necessary for this interaction.

14. Figure 4E: There seemed to be more UBQLN1 when UBQLN2 is knocked down, and more UBQLN2 when UBQLN1 is over-expressed. Does UBQLN1 or UBQLN2 regulate each other's expression/stability?

15. Figure 5C: The figure is cut off.

16. Figure 6: Since cell proliferation was not inhibited during the experiment, would it be possible that the wound closure was due to cell proliferation rather than migration, or the combination of these two?

17. Figure 7E: It would be helpful to show protein levels of EMT markers, such as SNAIL.

Minor editing of English language required. There are some redundant discussion of the results as well.

Author Response

We are thankful to reviewers for putting their time and effort towards enhancement of quality of manuscript. We really appreciate for the positive feedback regarding our manuscript and the work we carried out within the manuscript. According to reviewer’s suggestions, we have made numerous changes to the text within the manuscript and added additional experimental data further clarifying reviewer’s comments.

Please find below a comprehensive point-by-point clarification of all reviewer comments.

Comments and Suggestions for Authors

In this manuscript, the authors investigated the effect of UBQLN on tumor growth and metastasis and found that UBQLN regulates MYC levels in lung adenocarcinoma cells. It is an interesting topic to study, but the manuscript is poorly organized and riddled with mistakes and requires substantial proof-reading and reformatting. Some figures are cut off.

  1. Please provide methods for clonogenic assays, immunofluorescence microscopy, xenograft study in mice, and statistical analysis.

According to reviewer’s suggestion, details regarding methods for clonogenic assays, immunofluorescence microscopy, xenograft study in mice, and statistical analysis has been incorporated in the manuscript as a supplementary information.

  1. Figure 1A: Why did the NT control cells barely grow during the 5-day experiment period? Also, it would be great if the curves/datapoints can be made more distinguishable between conditions (i.e. different colors). Right now, it is hard to tell them apart.

We completely agree with the reviewer’s concern regarding NT control in Figure 1A. In this experiment, for comparison purpose all siRNA treatments for UBQLN1 and UBQLN2 knockdowns were first normalized to zero day reading and then normalized to NT control for each day, thus the NT controls are growing, but the difference is depicted as relative cell numbers compared to NT.  Curves/datapoints have been now made more distinguishable between conditions using different colors.  

  1. Figure 1C: Are the changes in cell cycle statistically significant? Maybe a histogram would be a better way to show the difference.

We completely agree with the reviewer’s concern regarding statistical significance in cell cycle. We did not see any statistical significance in case of cell cycle following loss of UBQLN1 and UBQLN2. To further confirm the possibility of cell proliferation, we carried out BrdU cell proliferation assay. As per reviewer’s suggestion histogram has been now incorporated in the figure.

  1. Figure 1D: Please specify which band is UBQLN1 or UBQLN2.

As per reviewer’s suggestion, specific bands for UBQLN1 or UBQLN2 has been now incorporated in the figure.

  1. Figure 1E: Please provide statistics for the differences in colony number. Also, does the difference in clonogenic potential between the two different siRNAs targeting the same gene correlate with the knockdown efficiency?

As per reviewer’s suggestion, statistics for the differences in colony number has been now incorporated in the figure. We completely agree with reviewer’s concern regarding knockdown efficiency for two different siRNAs targeting the same gene. Though, we always see 5 to 10% knockdown efficiency for two different siRNAs targeting the same gene, we always see overall consistence phenotypic changes including cell proliferation, cell migration and clonogenic potential.

  1. Figure S1A and S1B: siU2-6 has similar knockdown efficiency as siU2-5. Why did cells treated with siU2-6 form much fewer colonies than ones treated with siU2-5?

We completely agree with the reviewer’s concern regarding fewer colonies in case of siU2-6 compared to siU2-5 though the knockdown for both looks the same. For clonogenic assay, even though we plate equal number of cells to start with many times experimental conditions like washing of the cells, fixing, and staining play’s important role. We always try to avoid these kinds of errors by repeating experiments in triplicate conditions in three independent settings.  

  1. Figure S1C: It would be helpful to show the levels of these proteins in si_U2 cells as well.

We completely agree with the reviewer’s suggestion regarding levels of different proteins involved in cell cycle progression in case of si_U2 cells as well. As we performed these two experiments including clonogenic assay and western blot analysis in two different settings, we completely forgot to include si_U2 for western blot analysis. 

  1. Figure S2: The figure is cut off. Also, in S2Ai, there are 6 lanes on the western blot, but only 5 lanes are labeled.

We apologize for Figure S2 being cut off. Modified figure has been now incorporated in the manuscript. We highly appreciate reviewer for pointing out of mistake regarding labeling in figure S2Ai. Modified figure S2Ai has been now incorporated in the manuscript.

  1. Line 248 and 252: The text referred to the wrong figures.

We apologize for the incorrect statement regarding figures on line 248 and 252. The correct statement regarding proper figures has been now incorporated in the manuscript.

  1. Line 257: Figure S2Aiii does not exist.

We apologize for the statement regarding Figure S2Aiii. The correct statement has been now incorporated in the manuscript.

  1. Figure 2A: Please provide statistics and enrichment score.

The GSEA plot has been removed from the manuscript, as this data was generated some time ago and we were not able to find the statistics and score.  The removal of this panel does not alter the take home of the work.

  1. Figure 3: There seemed to be some delay in MYC degradation in Figure 3A, but it is hard to tell the kinetics in the remaining panels. Is there just more MYC to begin with? It would be important to calculate/measuring the actual half-life of MYC upon UBQLN knockdown.

We agree with the reviewer.  We have added to the discussion.

  1. Figure 4C: Does UBA also promote the interaction with MYC since the protein lacking UBA pulled down less MYC? Also, it would be helpful to do the same experiment with a mutant protein lacking UBL to show whether UBL is necessary for this interaction.

Yes, it appears that the interaction with MYC is UBL-dependent.

  1. Figure 4E: There seemed to be more UBQLN1 when UBQLN2 is knocked down, and more UBQLN2 when UBQLN1 is over-expressed. Does UBQLN1 or UBQLN2 regulate each other's expression/stability?

We completely agree with reviewers regarding expression levels of UBQLN1 and UBQLN2, when either one is loss. Currently, we are not completely sure whether UBQLN1 or UBQLN2 regulate each other's expression/stability. In fact, recently we published article demonstrating overlapping and distinct roles of UBQLN1 and UBQLN2 in lung cancer progression and metastasis. (Shah et al., Neoplasia 2022). In this manuscript, we provided evidence that UBQLN1 and UBQLN2 interacts with each other and have overlapping molecular and cellular functions. Future study will definitely focus on in depth analysis of UBQLN1 and UBQLN2 expression and stability.

  1. Figure 5C: The figure is cut off.

We apologize for Figure 5C being cut off. Figure in the proper format has been now incorporated in the manuscript.

  1. Figure 6: Since cell proliferation was not inhibited during the experiment, would it be possible that the wound closure was due to cell proliferation rather than migration, or the combination of these two?

We really thank reviewer’s for raising important question regarding cell migration assay. According to reviewer’s suggestion, in order to clarify the wound closure in case of UBQLN loss was really due to migration and mobility of cells and not due to increase in cell proliferation, we performed single cell time lapse mobility assay for 24 hrs using Keyence live cell imager. Interestingly, we observed significant increase in cell mobility following loss of UBQLN1 compared to cells transfected with non-targeting control siRNA. We further noticed that there was not significant increase in cell proliferation. This data clearly demonstrates that increase in cell migration in case lung adenocarcinoma cells following loss of UBQLN1 was due to increase in cell mobility and not due to increase in cell proliferation.

  1. Figure 7E: It would be helpful to show protein levels of EMT markers, such as SNAIL.

According to reviewer’s suggestion, we have incorporated western blot showing the protein level of EMT markers such as Snail. We further investigated expression level of key proteins including E-cadherin, Claudin1, N-cadherin, Slug, Zeb1 and β-catenin which are known for their important role in regulation of EMT.

Round 2

Reviewer 1 Report

The authors have adequately addressed my concerns

Author Response

Thank you

Reviewer 2 Report

The authors have addressed most of the comments. 

Minor:

Figure 5C, X axis, second and third labels to the right: is NT included in tne label by mistake?

English is fine. Some reformatting is needed

Author Response

We have fixed figure 5. thank you.